# Epistatic Net allows the sparse spectral regularization of deep neural networks for inferring fitness functions

Amirali Aghazadeh [1], Hunter Nisonoff[2], Orhan Ocal[1], David H. Brookes[3], Yijie Huang[1], O. Ozan Koyluoglu[1], Jennifer Listgarten [1,2] & Kannan Ramchandran [1✉]

Despite recent advances in high-throughput combinatorial mutagenesis assays, the number of labeled sequences available to predict molecular functions has remained small for the vastness of the sequence space combined with the ruggedness of many fitness functions. While deep neural networks (DNNs) can capture high-order epistatic interactions among the mutational sites, they tend to overfit to the small number of labeled sequences available for training. Here, we developed Epistatic Net (EN), a method for spectral regularization of DNNs that exploits evidence that epistatic interactions in many fitness functions are sparse. We built a scalable extension of EN, usable for larger sequences, which enables spectral regularization using fast sparse recovery algorithms informed by coding theory. Results on several biological landscapes show that EN consistently improves the prediction accuracy of DNNs and enables them to outperform competing models which assume other priors. EN estimates the higher-order epistatic interactions of DNNs trained on massive sequence spaces-a computational problem that otherwise takes years to solve.

[1] Department of Electrical Engineering and Computer Sciences, Berkeley, CA, USA. [2] Center for Computational Biology, Berkeley, CA, USA. [3] Biophysics Graduate Group, University of California, Berkeley, CA, USA. ✉email: kannanr@eecs.berkeley.edu

Recent advances in next-generation sequencing have enabled the design of high-throughput combinatorial mutagenesis assays that measure molecular functionality for tens of thousands to millions of sequences simultaneously. These assays have been applied to many different sequences in biology, including protein-coding sequences[1–3], RNAs[4–6], bacterial genes[7–10], and the Cas9 target sites[11–13]. The labeled sequences collected from these assays have been used to train supervised machine learning (ML) models to predict functions (e.g., fluorescence, binding, repair outcome, etc.) from the sequence—a key step in the rational design of molecules using ML-assisted directed evolution[14]. However, due to the limitations in techniques for library preparation, these assays can only uncover a small subset of all the possible combinatorial sequences. This raises an important question in learning fitness functions: how can we enable supervised ML models to infer fitness functions using only a small number of labeled sequences?

Inferring fitness functions is a challenging task since mutational sites interact nonlinearly to form the function, a phenomenon known as epistasis in genetics[15,16]. As a result, linear regression models which assume site-independent interactions achieve poor accuracy in predicting nonlinear functions. Augmenting linear models with pairwise, second-order epistatic interactions improves their prediction accuracy[3]; however, there is now increasing evidence that a large fraction of the variance in the fitness functions can be explained only by higher-order epistatic interactions, which contribute to the ruggedness of fitness landscapes[17,18]. Modeling rugged fitness landscapes is a hard task since the total number of possible higher-order interactions grows exponentially with the number of mutational sites. As a result, the number of parameters to be estimated (i.e., the problem dimension) also grows with the same exponential rate, which creates statistical challenges in inferring the fitness function since the number of labeled sequences does not scale with the problem dimension. In response, nonlinear ML models constrain the problem dimension by introducing various forms of inductive biases to capture hidden structures in the fitness functions. Random forests, for example, impose a tree structure over sites which favor tree-like hierarchical epistatic interactions. While these inductive biases are effective in some fitness functions[19], they are too restrictive to capture the underlying higher-order epistatic interactions in other fitness functions[3]. Over-parameterized models in deep learning (DL), such as deep neural networks (DNNs), are expressive enough to model high-order epistatic interactions given a large number of labeled training sequences; however, when the number of labeled sequences is small, they often overfit to the training data and compromise prediction accuracy. It has been observed that regularizing DNNs to induce domain-specific biases improves their prediction accuracy for various tasks in computer vision and natural language processing[20]. This opens up the question of whether there exists an inductive bias for DNNs trained on biological fitness landscapes that can be imposed using a computationally tractable regularization scheme.

Recent studies in biological landscapes[3,13,21] have reported that a large fraction of the variance in many fitness functions can be explained by only a few number of (high-order) interactions between the mutational sites. The epistatic interactions in these functions are a mixture of a small number of interactions with large coefficients, and a larger number of interactions with small coefficients; in other words, their epistatic interactions are highly sparse. Promoting sparsity among epistatic interactions is a powerful inductive bias for predictive modeling because it reduces the problem dimension without biasing the model towards a subset of (low-order) interactions. Despite its benefits, promoting sparsity among epistatic interactions has not been studied in

DNNs as an inductive bias. The roadblock is in finding a method to promote epistatic sparsity in DNNs. Unfortunately, directly penalizing all or some of the parameters (weights) of DNNs with sparsity-promoting priors is not likely to result in sparse epistatic regularization since the epistatic coefficients are a complex non-linear function of the weights in DNNs.

Here, we develop a method for sparse epistatic regularization of DNNs. We call our method *Epistatic Net* (EN) because it resembles a fishing net which catches the epistatic interactions among all the combinatorially possible interactions in DNNs, without any restriction to a subset of (low-order) interactions. In order to find the epistatic interaction as a function of the weights in DNN, we find its spectral representation (also called the Walsh-Hadamard (WH) transform for binary sequences) by evaluating the DNN on the entire combinatorial space of mutations, and then take the WH spectral transform of the resulting landscape using the Fast WH Transform (FWHT). The resulting function of the weights in DNN is penalized to promote epistatic sparsity. For larger sequences this approach for epistatic regularization becomes computationally intractable due to the need to enumerate all possible mutations in DNN. Therefore, we leverage the fast sparsity-enabled algorithms in signal processing and coding theory in order to develop a greedy optimization method to regularize DNNs at scale. Our scalable regularization method, called EN-S, regularizes DNNs by sampling only a small subset of the combinatorial sequence space by choosing sequences that induce a specific sparse graph structure. The uniform sampling scheme allows us to find the WH transform of the combinatorial DNN landscape efficiently using a fast peeling algorithm over the induced sparse graph[22]. Results on several biological landscapes, from bacterial to protein fitness functions, shows that EN(-S) enables DNNs to achieve consistently higher prediction accuracy compared to competing models and estimate all the higher-order predictive interactions on massive combinatorial sequence space—a computational problem that takes years to solve without leveraging the epistatic sparsity structure in the fitness landscapes.

## Results

**Regularization using the Epistatic Net (EN)**. EN is a novel regularization scheme (Fig. 1b) which evaluates the DNN on all the possible combinatorial mutations of the input sequence; we call the resulting high-dimensional vector the DNN landscape. EN takes the WH transform of the DNN landscape and adds the sparsity-promoting $\ell_1$-norm (i.e., the sum of the absolute values) of the WH coefficients (or total sum of the magnitude of epistasis) to the log-likelihood loss. The resulting WH loss is a differentiable function (except at zero) of the weights in DNN and is weighted by a scalar which strikes a balance between the fidelity of DNN to the labeled sequences and sparsity among epistatic interactions (see "Methods", Supplementary Notes, and Supplementary Fig. 1). We use the stochastic gradient descent (SGD) algorithm to minimize the aggregate loss and update the weights of DNN in every iteration.

For larger sequences (of size $d > 25$), EN regularization becomes intractable in time and space complexity. This is because EN needs to query the DNN $p = 2^d$ times to form the DNN landscape (exponential time complexity in $d$) and then find the WH transform of the queried DNN landscape (exponential time and space complexity in $d$). To overcome this, EN-S leverages the sparsity in the WH spectral domain to regularize DNN using only a small number of uniformly subsampled sequences from the combinatorial input space (Fig. 1c). EN-S decouples the DNN training, following the alternating direction method of multipliers (ADMM) framework[23], into two

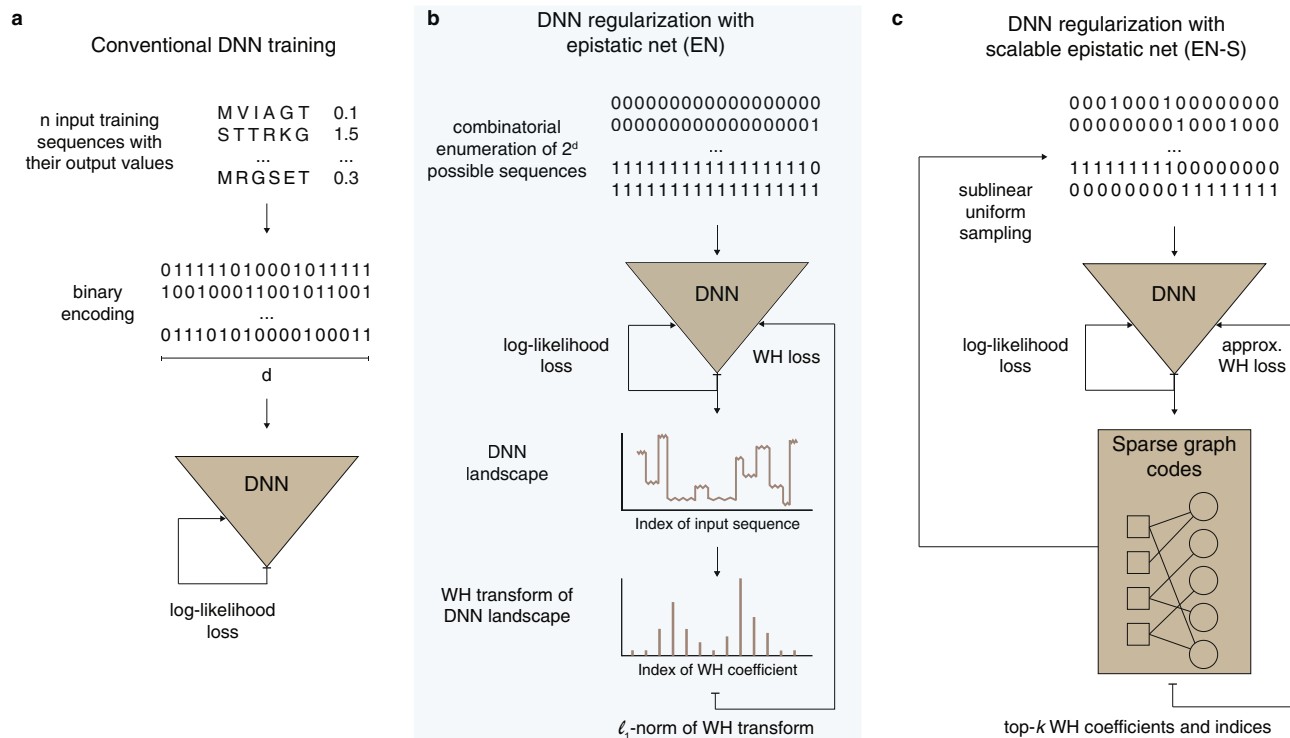

**Fig. 1 Schematic illustration of our sparse epistatic regularization method, called Epistatic Net (EN). a** Conventional deep neural network (DNN) training is depicted, where the log-likelihood loss (computed over $n$ labeled training sequences encoded into binary sequences of length $d$) is minimized using the stochastic gradient descent (SGD) algorithm. **b** In every iteration, EN queries DNN for all the $2^d$ possible binary input sequences, finds the Walsh Hadamard (WH) spectral transform of the resulting landscape using the Fast WH Transform (FWHT), and then adds the $\ell_1$-norm of the WH transform to the log-likelihood loss from panel (**a**). **c** In the scalable version of EN, EN-S regularizes DNN using only a few number of uniformly subsampled sequences from the combinatorial input space that casts the sparse WH recovery problem on an induced sparse-graph code. EN-S iterates between these two subproblems until convergence: (1) finding the sparse WH transform of DNN (using sublinear samples and in sublinear time) through peeling over the induced sparse-graph codes, and (2) minimizing the sum of the log-likelihood loss and the WH loss using SGD.

subproblems: (1) finding the $k$-sparse WH spectral transform of DNN in a sample and time efficient manner, and (2) minimizing the sum of the log-likelihood loss and the WH loss. The WH loss penalizes the distance between DNN and a function constructed using the top-$k$ WH coefficients recovered in the first subproblem. In order to solve the first subproblem, we design a careful subsampling of the input sequence space[22] that induces a linear mixing of the WH coefficients such that a greedy belief propagation algorithm (peeling-decoding) over a sparse-graph code recovers the noisy DNN landscape in sublinear sample (i.e., $\mathcal{O}(k\log^2 p)$) and time (i.e., $\mathcal{O}(k\log^3 p)$) complexity in $p$ (with high probability)[13,22,24,25]. Briefly, the peeling-decoding algorithm identifies the nodes on the induced sparse-graph code that are connected to only a single WH coefficient and peels off the edges connected to those nodes and their contributions on the overall graph. The algorithm repeats these steps until all the edges are removed. We solve the second subproblem using the SGD algorithm. EN-S alternates between these two steps until convergence (see "Methods" and Supplementary Notes).

**Inferring four canonical functions in bacterial fitness.** We collected four canonical bacterial fitness functions, whose combinatorial landscapes have been measured experimentally in previously published works (see Supplementary Table 1). Figure 2a shows the sparsity level in epistatic interactions of these bacterial fitness functions. We found the coefficients for epistatic interactions by taking the WH transform of the measured combinatorial landscape (see "Methods" section for various ways to preprocess the landscapes). Figure 2a plots the fraction of

variance explained as a function of the top WH coefficients. Sparsity levels can be assessed by the proximity of the resulting curve towards the top-left corner of the plot. For comparison, we also plotted synthetic fitness functions that have all possible epistatic interactions up to a certain order of interactions in Fig. 2a. While the sparsity levels vary across fitness functions, the top-5 WH coefficients consistently explain more than 80% of the variance across all the landscapes.

Figure 2b shows the prediction performance of DNN with EN regularization on the bacterial landscapes compared to various competing models. All the models are trained on the same randomly sampled subset (i.e., 31%) of the sequences from the measured combinatorial landscapes and tested on a subset of unseen sequences (see Supplementary Notes for more details). The prediction accuracy is reported in terms of the coefficient of determination, $R^2$ (i.e., the fraction of the variance in the test set explained from the sequence). DNN with EN regularization consistently outperforms the baseline models in all the landscapes. In particular, DNN with EN regularization performs significantly better than the EN-unregularized variant consistently across all data sets ($\Delta R^2 > 0.1$, $P < 0.033$), even though DNN is optimized (in terms of architecture) for best validation performance in isolation (i.e., without epistatic regularization) and has been subjected to other forms of common sparsity-promoting regularization techniques applied directly to the weights of the DNN ("Methods", Supplementary Data 1, and Supplementary Fig. 2).

Figure 2c shows the WH transform of the DNN landscape with and without EN regularization, as well as the WH transform of the landscapes corresponding to the rest of the competing models

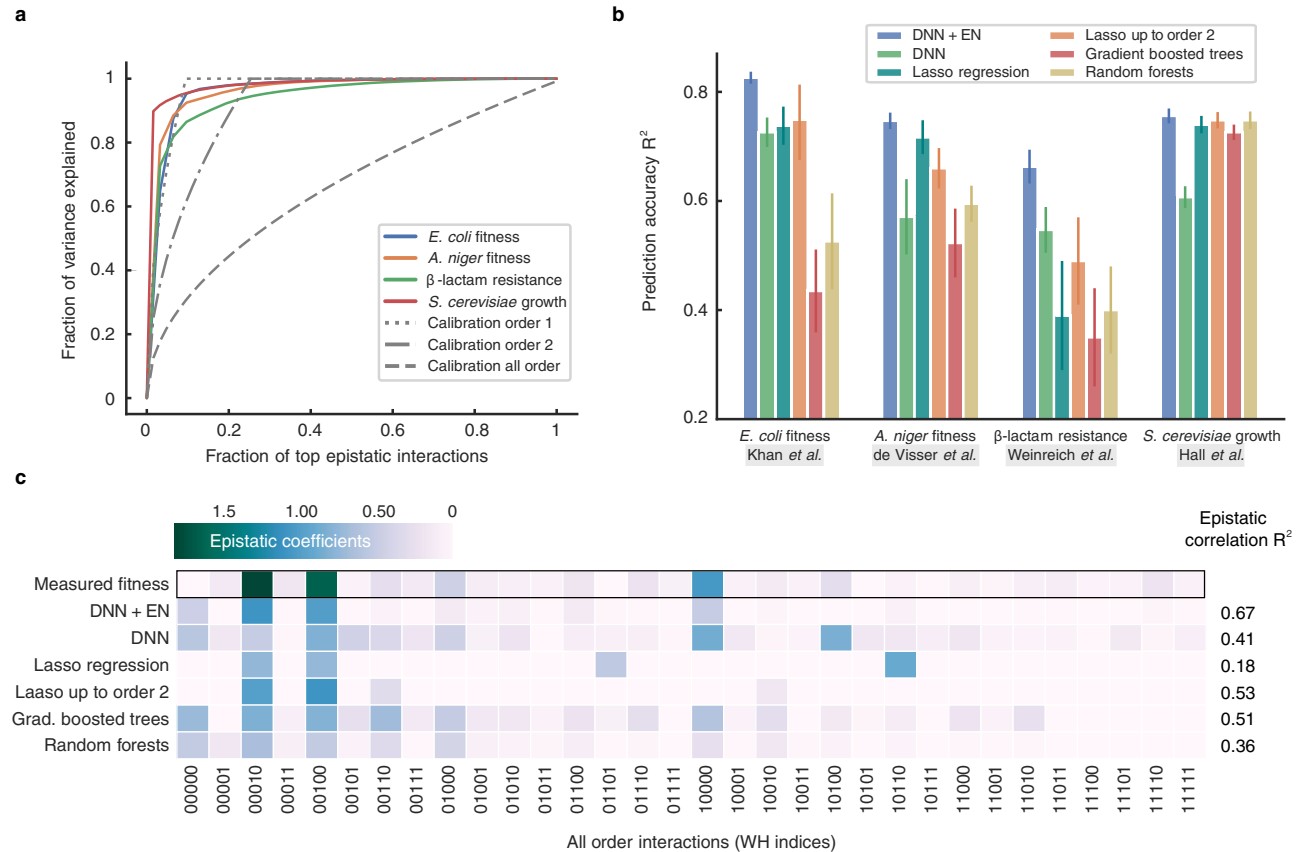

**Fig. 2 Predicting bacterial fitness and inferring epistatic interactions in four canonical landscapes. a** Fraction of variance explained by the top WH coefficients revealing the sparsity in the bacterial fitness functions. **b** Prediction accuracy of deep neural network (DNN) with Epistatic Net (EN) regularization against competing models in ML. The error bars show the standard error of the mean (SEM) across 20 independent repeats of the experiments with random split of the data into training, validation, and test sets. **c** Visualization of the epistatic interactions of DNN with and without EN regularization and the baseline models after training on *Escherichia coli* fitness landscape of Khan et al.[9]. $R^2$ values show the correlation of the recovered epistatic interaction with the interactions in the measured combinatorial *Escherichia coli* fitness landscape.

trained on a training set sampled from the *Escherichia coli* fitness landscape of Khan et al.[9] (see Supplementary Figs. 3, 4 for a detailed analysis of the landscapes in the spectral domain). In order to find these landscapes, we queried each model for all the combinatorial mutations. In this plot, the epistatic coefficient indexed by 10100, as an example, shows an order 2 interaction between the mutational sites 1 and 3. The rest of the indices can be interpreted similarly. The WH coefficients in the measured *Escherichia coli* fitness function show three first-order interactions with higher magnitude and several higher-order interactions with lower magnitude. The interactions recovered by DNN with EN regularization closely match the epistatic interactions of the measured *Escherichia coli* fitness function ($R^2 = 0.67$), a considerable improvement over DNN without EN regularization ($R^2 = 0.41$). EN regularization effectively denoises the WH spectrum of DNN by removing spurious higher-order interactions; nevertheless, given a larger training set, EN would have accepted a larger number of higher-order interactions. The WH coefficients of gradient boosted trees ($R^2 = 0.51$) and random forests ($R^2 = 0.36$) also show several spurious high-order interactions. Lasso regression finds two of the three measured interactions with higher magnitude, however, recovers a spurious third-order interaction which results in a low epistatic correlation coefficient ($R^2 = 0.18$). When restricted to up to order 2 interactions, the performance of Lasso improves; it recovers the two interactions with higher coefficients, however, misses the third coefficient and the rest of the small epistatic interactions ($R^2 = 0.53$).

**Entacmaea quadricolor fluorescent protein**. A comprehensive experimental study has reported all the combinatorial mutants that link two phenotypically distinct variants of the *Entacmaea quadricolor* fluorescent protein[3]. The variants are different in $d = 13$ mutational sites. The study shows the existence of several high-order epistatic interactions between the sites, but also reveals extraordinary sparsity in the interactions. We used this protein landscape to assess EN in regularizing DNN for predicting protein function. We split the $2^{13} = 8192$ labeled proteins randomly into three sets: training, validation, and test. The size of the test set was fixed to 3000 and the validation set size was set equal to the training set size. We varied the training set size from a minimum of $n = 20$ proteins to a maximum of $n = 100$ proteins and evaluated the accuracy of the models in (1) predicting fitness in Fig. 3a in terms of $R^2$ (Supplementary Data 2) and (2) recovering the experimentally measured epistatic interactions in Fig. 3b in terms of normalized mean squared error (NMSE) (Supplementary Data 3).

DNN with EN regularization significantly outperforms DNN without EN regularization in terms of prediction accuracy ($\Delta R^2 > 0.1$, $P < 10^{-5}$), consistently across all training sizes. Moreover, DNN with EN regularization recovers the experimentally measured epistatic interactions with significantly lower error ($\Delta$NMSE $> 0.07$, $P < 9 \times 10^{-5}$), consistently across all training sizes. Applying various forms of $\ell_1$ and $\ell_2$-norm regularization on the weights of different layers of the DNN does not change the performance gap between DNN with and without EN regularization (see Supplementary Fig. 5). In particular, in order to achieve

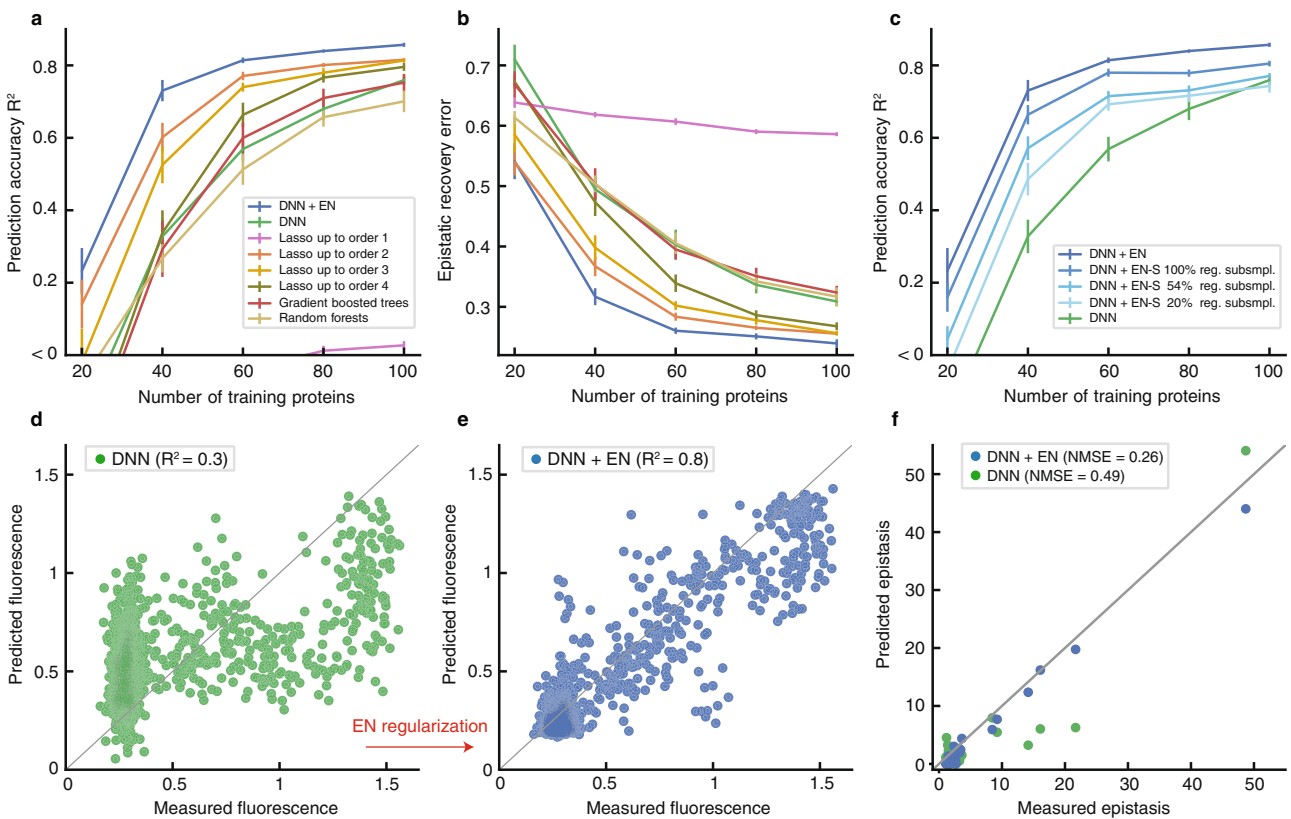

**Fig. 3 Inferring the sparse epistatic protein landscape of Poelwijk et al.[3]. a** Deep neural network (DNN) with Epistatic Net (EN) regularization outperforms the baselines in terms of prediction accuracy. To achieve the same prediction accuracy, DNN with EN regularization needs up to 3 times less number of samples compared to DNN without EN regularization. **b** DNN with EN regularization recovers the experimentally measured (higher-order) epistatic interactions with significantly lower normalized mean squared error (NMSE). **c** The prediction performance of DNN with EN-S regularization is plotted when EN-S subsamples DNN at progressively smaller fractions of the combinatorial sequence space of proteins, that is, 100% (no subsampling), 54%, and 20%. DNN with EN-S regularization outperforms DNN without the regularization despite restricting EN-S to only sample 20% of the protein sequence space to induce a sparse-graph code. Error bars in all the plots show the standard error of the mean (SEM) in 20 independent repeats of the experiments with random splits of the data into training, validation, and test sets. **d** Scatter plot of the DNN-predicted fluorescence values trained on $n = 60$ labeled proteins. **e** Scatter plot of the predicted fluorescence values by the EN-regularized variant of the same DNN. **f** Comparison of the recovered epistatic interactions of the EN-regularized and unregularized DNNs.

the same level of prediction accuracy ($R^2 = 0.7$), DNN without EN regularization requires up to 3 times more training samples compared to DNN with EN regularization. Figures 3d, e show the scatter plots of the predicted fluorescence values of DNN and its EN-regularized variant, respectively, when both models are trained on $n = 60$ labeled proteins. The performance gap naturally reduces for larger training sets, however, it stays consistently positive even up to $n = 200$ (i.e., 2.5% of the entire combinatorial landscape), which is typically larger than the number of available labeled sequences in protein function prediction problems (Supplementary Fig. 5). Our analysis also reveals the improved performance of the epistatic interactions recovered by DNN with EN regularization in predicting the pairwise contacts (residues with smaller than 4.5 Å distance[26]) and triplet contacts (group of three residues with smaller than 4.5 Å pairwise distances) in the 3D structure of the protein—even though the networks are not trained for protein structure prediction task (Supplementary Fig. 6). DNN with EN regularization predicts contacts with $F_1^{order\ 2} = 0.76$ and $F_1^{order\ 3} = 0.68$ compared to DNN without EN regularization with $F_1^{order\ 2} = 0.67$ and $F_1^{order\ 3} = 0.66$ ($F_1$ score takes the harmonic mean of the precision and recall rates).

The dimension of the fluorescent landscape of *Entacmaea quadricolor* protein enabled us to use the data set to compare the

performance of DNN under EN regularization with its scalable version, EN-S. The prediction performance of DNN with EN-S regularization showed a slight drop in accuracy due to the approximations made by the ADMM decoupling (Fig. 3c, Supplementary Data 4, and "Methods"). EN-S stayed fairly consistent when we decreased the number of proteins sampled from DNN to induce a sparse-graph code. Using as low as 1678 samples (out of the total of 8192 combinatorial proteins, i.e., 20% subsampling) enabled successful regularization of DNN, resulting in a significant performance gap compared to DNN without EN regularization.

**Green fluorescent protein from *Aequorea victoria* (avGFP).** The local fitness landscape of the green fluorescent protein from *Aequorea victoria* (avGFP) has been investigated in a comprehensive study[2]. The authors estimated the fluorescence levels of genotypes obtained by random mutagenesis of the avGFP protein sequence at 236 amino acid mutational sites. The final data set included 56,086 unique nucleotide sequences coding for 51,715 different protein sequences. Considering the absence or presence of a mutation at a site, created a data set with input sequence size of $d = 236$. Regularization in the resulting $p = 2^{236}$-dimensional space was impossible using EN, illustrating the need for EN-S. We first analyzed the peeling algorithm by inspecting the WH

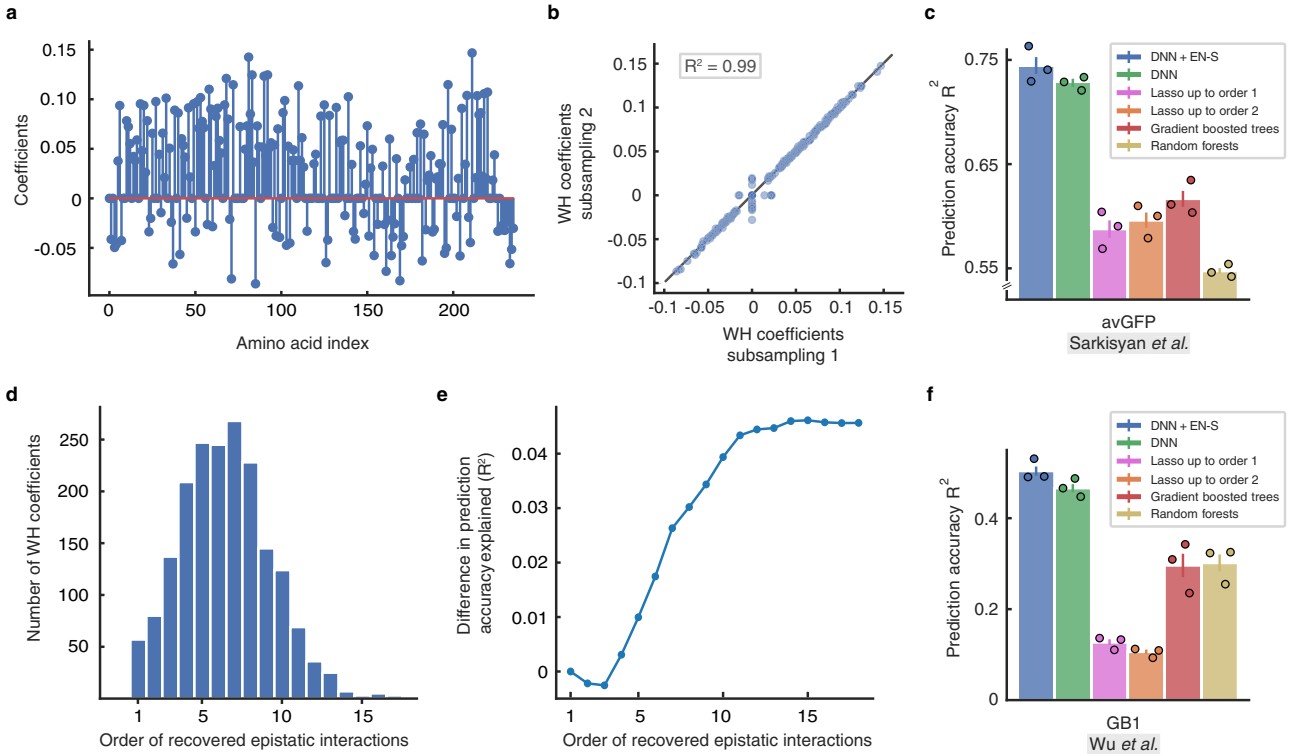

**Fig. 4 Inferring epistatic interactions in two large canonical protein landscapes using the scalable Epistatic Net (EN-S) regularizer. a** The first-order Walsh-Hadamard (WH) coefficients of unregularized DNN trained on the *Aequorea victoria* (avGFP) landscape of Sarkisyan et al.[2] recovered by the peeling algorithm using a set of $5,074,944$ uniformly subsampled proteins (out of $2^{236}$). **b** The scatter plot of the first-order WH recovered by EN-S using two independent sets of $5,074,944$ uniformly subsampled proteins. The recovered coefficients are highly consistent ($R^2 = 0.99$). The higher variance of the scatter plot around the center shows the small number (20 out of 236) of coefficients that are differentially recovered. **c** DNN with EN-S regularization outperforms the baselines in terms of prediction accuracy in avGFP. **d** Histogram of the order of epistatic interactions recovered while training the EN-S regularized DNN. **e** The prediction accuracy gained by the higher-order epistatic interactions when added to a purely linear model. **f** DNN with EN-S regularization outperforms the baselines in terms of prediction accuracy in the GB1 landscape of Wu et al.[1]. Error bars show the standard error of the mean (SEM) in 3 independent repeats of the experiments with random splits of the data into training, validation, and test sets.

spectral representation of DNN once trained on the avGFP landscape. Figure 4a shows the first-order WH coefficients of DNN, recovered using peeling after sampling DNN at 5,074,944 (out of $2^{236} \approx 10^{71}$) proteins following uniform patterns that induce a sparse-graph code. We repeated the same procedure with an independent set of uniformly subsampled sequences (with random offset) and visualized the recovered first-order WH coefficients in a scatter plot as a function of the recovered coefficients using the first set of proteins in Fig. 4b. When sampled at two different relatively tiny subsets of this massive $p = 2^{236}$-dimensional space, the peeling algorithm recovered similar first-order coefficients (with $R^2 = 0.99$), without assuming any prior knowledge on the WH coefficients of avGFP being low-order (also see Supplementary Fig. 7). The higher variance of the scatter plot around the center shows the small number of coefficients (30 out of 236) that are differentially recovered under the two subsamplings. The peeling algorithm associated 3.2% and 2.9% of the variation of DNN to higher-order interactions, respectively for the first and second subsampling. We compared the second-order interactions recovered under these subsamplings (Supplementary Fig. 8). Despite the small variation associated with higher-order epistasis, 10% of the recovered second-order interactions were exactly equal, and the rest of the interactions were locally correlated ($R^2 = 0.60$ correlation within blocks of three neighboring interactions).

Next, we trained the same DNN architecture with EN-S regularization. Figure 4c shows that the prediction accuracy of DNN with EN-S regularization is higher than the baseline

algorithms (Supplementary Data 5). The gap between DNN with and without EN-S regularization is smaller compared to the previously described protein landscapes. We speculate that this is due to the nature of the local landscape of avGFP around the wild-type protein, where most of the variance can be explained by first-order interactions and the rest can be explained by higher-order interactions that are spread throughout the WH spectrum. Figure 4d illustrates the histogram of the order of epistatic interactions recovered by invoking the peeling algorithm in every iteration of the EN-S regularization scheme. Figure 4e depicts the gain in prediction accuracy after adding the recovered interactions to a purely linear model, suggesting that the difference in prediction accuracy of DNN with and without regularization can be explained (approximately) by a collection of a large number of WH coefficients with small magnitude—this analysis further demonstrates the computational power of EN-S in recovering higher-order interactions in such massively large combinatorial space of interactions.

**Immunoglobulin-binding domain of protein G (GB1).** A recent study investigated the fitness landscape of all the $20^4 = 160,000$ variants at four amino acid sites (V39, D40, G41, and V54) in an epistatic region of protein G domain B1, an immunoglobulin-binding protein expressed in Streptococcal bacteria[1]. One-hot binary encoding of the amino acids results in binary sequences of length $d = 80$. As EN does not scale to regularize DNNs trained on this landscape, we relied on EN-S. Figure 4f shows the prediction performance of DNN with EN-S regularization compared

to the baseline models that were scaled to such a dimension. All the models were trained on a random subset of $n = 2000$ proteins. EN-S subsamples DNN at 215,040 proteins in order to perform the sparse epistatic regularization, which is about $10^{18}$ times smaller than the entire sequence space. Despite such an enormous level of undersampling, the DNN regularized with EN-S consistently outperforms the competing baselines and the EN-S unregularized DNN ($\Delta R^2 > 0.035$, $P < 0.05$, Supplementary Data 5, and Supplementary Fig. 9). The performance gap between the DNNs with and without EN-S regularization is naturally smaller compared to the same gap in the *Entacmaea quadricolor* fluorescent protein landscape. This is because the protein landscape of *Entacmaea quadricolor* is defined over 13 mutational sites (with 8192 possible positional interactions and two possible amino acids for each site) while the protein landscape of GB1 is defined over 4 mutational sites (with 16 possible positional interactions and 20 possible amino acids for each site); the former benefits more from promoting sparsity among a larger number of biologically meaningful positional interactions.

## Discussion

We showed that several of the functional landscapes in biology have common structures (i.e., inductive bias) in their epistatic interactions that manifest as sparsity in the spectral Walsh–Hadamard (WH) domain. Sparse epistatic regularization of deep neural networks (DNNs) is an effective method to improve their prediction accuracy, especially when the number of available training samples is small compared to the vastness of sequence space. To this end, our Epistatic Net (EN) regularization method combined the advantages offered by the sparsity of biological landscapes with sublinear algorithms in signal processing and coding theory for epistatic regularization of DNNs in the combinatorial space of interactions. Analysis of the recovered higher-order epistatic interactions by the DNNs with and without regularization also revealed the power of EN in finding biologically relevant epistatic interactions.

The superior prediction performance of DNNs with EN regularization comes with the additional computational cost of finding the WH transform of the DNN landscape, which increases the computational complexity of the training algorithm by only a linear factor in the product of the length of the sequence and the epistatic sparsity level. While training can be done offline (e.g., on a server) there are avenues for making the algorithm even more efficient such as using the prior knowledge on the maximum order of interaction to constrain the regularization space. In addition, EN regularization can be extended using generalized Fourier transform to more efficiently encode amino acids compared to the more conventional one-hot binary encoding strategies[27]. Moreover, while this work laid out the algorithmic principles of sparse epistatic regularization in supervised models, unsupervised models, such as Potts model[28], Ising model[29], and Variational Autoencoders (VAEs)[30] can benefit from such regularization scheme as well; it would be tempting to hypothesize that these energy landscapes also have structures that appear as high-order sparse coefficients in WH basis.

Overall, our sparse epistatic regularization method expands the machine learning toolkit for inferring and understanding fitness functions in biology. It helps us to visualize, analyze, and regularize the powerful, however less interpretable black-box models in deep learning in terms of their higher-order interactions in the sequence space. We believe that our work will initiate new research directions towards developing hybrid methodologies that draws power from statistical learning, signal processing, coding theory, and physics-inspired deep learning for protein design and engineering.

## Methods

**Notation and background**. Suppose we are given $n$ (experimental) samples $(\mathbf{x}_i, y_i)_{i=1}^n$, that is, (sequence, value) pairs from a biological landscape, where $\mathbf{x}_i \in \{-1, +1\}^d$ denotes the binary encoding of $d$ mutational sites in a variant and $y_i \in \mathbb{R}$ is its associated fitness value. We are interested in learning a function $f(\mathbf{x})$ that maps all subsets of mutations to fitness values. In other words, we seek to learn a set function $f(\mathbf{x}) : \mathbb{F}^d \to \mathbb{R}$, where $\mathbb{F}^d$ denotes the space of all the binary vectors of length $d$. A key theorem[31] in mathematics states that any set function (also known as pseudo-Boolean function) $f(\mathbf{x}) = f(x_1, x_2, \ldots, x_d)$ can be represented uniquely by a multi-linear polynomial over the hypercube $(x_1, x_2, \ldots, x_d) \in \{-1, +1\}^d$:

$$f(x_1, x_2, \ldots, x_d) = \sum_{\mathcal{S} \subseteq [d]} \alpha_{\mathcal{S}} \prod_{i \in \mathcal{S}} x_i, \tag{1}$$

where $\mathcal{S}$ is a subset of $\{1, 2, 3, \ldots, d\} = [d]$ and $\alpha_{\mathcal{S}} \in \mathbb{R}$ is the WH transform coefficient (or equivalently the epistatic coefficient) associated with the monomial (interaction) $\prod_{i \in \mathcal{S}} x_i$. For example, the pseudo-Boolean function

$$f(x_1, x_2, x_3, x_4, x_5) = 12x_1 x_4 - 3x_3 + 6x_1 x_2 x_5, \tag{2}$$

defined over $d = 5$ mutational sites, has three monomials with orders 2, 1, and 3 and WH coefficients 12, $-3$, and 6, respectively. The WH transform of this function is sparse with $k = 3$ non-zero coefficients out of a total of $2^5 = 32$ coefficients. Each monomial can be easily explained, for example, the first monomial in the WH transform, that is $12x_1 x_4$, indicates that mutation sites 1 and 4 are interacting and the interaction enriches fitness because the sign of the coefficient is positive. On the hand, the second monomial $-3x_3$ shows that a mutation at site 3 depletes fitness. The last monomial $6x_1 x_2 x_5$ shows a third-order interaction between mutational sites 1, 2, and 5 which also enrich fitness.

If the fitness function is measured (known) for all the combinatorial $p = 2^d$ inputs $\mathbf{x}_i$, then we can use the Fast WH Transform (FWHT)[32] to find the WH coefficients in $\mathcal{O}(p \log p)$ time complexity. The problem is so-called fully determined in such a scenario. However, as discussed in the introduction, in inferring fitness functions, we typically face problems where the number of observed samples (sequences) $n$ is much smaller than the total number of possible sequences, that is, $n \ll p = 2^d$; in other words, we are in an underdetermined regime. In full generality, we assume that the data is generated according to a noisy nonlinear model

$$y_i = f_\theta(\mathbf{x}_i) + \varepsilon_e, \tag{3}$$

where $\theta$ are the parameters of the model, $\varepsilon_e$ is a random variable drawn from a Gaussian distribution with zero mean and variance $\sigma_e^2$. Under this setting the maximum likelihood estimate is

$$\theta_{MLE} = \arg \max_\theta \sum_{i=1}^n (y_i - f_\theta(\mathbf{x}_i))^2. \tag{4}$$

We denote a deep neural network (DNN) by $g_\theta(\mathbf{x})$, where $\theta$ is a vector of all the weights in DNN. The DNN, $g_\theta(\mathbf{x})$, takes in a binary input vector $\mathbf{x}_i$ and predicts the output $\hat{y}_i$. Let $\mathbf{X} \in \mathbb{R}^{p \times d}$ denote a matrix which comprises all the $p = 2^d$ enumeration of the binary sequence $\mathbf{x}_i$ of length $d$ in its rows. We slightly abuse the notation and let $\mathbf{g}_\theta(\mathbf{X}) \in \mathbb{R}^p$ denote the real-valued vector of DNN outputs over all these binary sequences. We call this high-dimensional vector the DNN landscape. In order to find the WH transform of the DNN we can multiply the DNN landscape, $\mathbf{g}_\theta(\mathbf{X})$, by the WH matrix, $\mathbf{H} \in \mathbb{R}^{p \times p}$. The WH matrix $\mathbf{H}$ can be defined using the recursive equation

$$\mathbf{H}^{2^d} = \mathbf{H}^2 \otimes \mathbf{H}^{2^{d-1}}, \tag{5}$$

where $\mathbf{H}^2$ is the $2 \times 2$ mother WH matrix defined as $\mathbf{H}^2 = \begin{bmatrix} 1 & 1 \\ 1 & -1 \end{bmatrix}$ and $\otimes$ denotes the Kronecker product. The WH matrix is a symmetric unitary matrix; in other words, $(1/2^d)\mathbf{H}\mathbf{H} = \mathbf{I}$. Each of the $2^d$ columns of $\mathbf{H}$ corresponds to a monomial ($\prod_{i \in \mathcal{S}} x_i$) in the pseudo-Boolean representation of set functions and equivalently corresponds to one of the terms in WH transform. In biology literature, this coefficient is known as an epistatic interaction when $|\mathcal{S}| \geq 2$. The WH transform of the DNN can be calculated as $\mathbf{H}\mathbf{g}_\theta(\mathbf{X}) \in \mathbb{R}^p$. Note that in this manuscript we assume $\mathbf{H}$ is properly normalized to be unitary.

**Epistatic net (EN)**. EN regularizes the epistatic interactions in $\mathbf{g}_\theta(\mathbf{X})$ by adding a new WH loss term to the original log-likelihood loss,

$$\min_\theta \sum_{i=1}^n (y_i - g_\theta(\mathbf{x}_i))^2 + \alpha \|\mathbf{H}\mathbf{g}_\theta(\mathbf{X})\|_0, \tag{6}$$

where $\mathbf{H} \in \mathbb{R}^{p \times p}$ is the WH matrix, the $\ell_0$-norm $\|.\|_0$ counts the number of non-zero values in the WH transform of the DNN (i.e., $\mathbf{H}\mathbf{g}_\theta(\mathbf{X})$), and $\alpha$ is a scalar which strikes balance between the log-likelihood loss and the regularization term. The scalar $\alpha$ is set using cross-validation. The $\ell_0$-norm is a non-convex and non-differentiable term and is not suitable for optimization using the SGD algorithm since the gradient is not well-defined for this term; therefore, following the common practice in convex optimization, we relaxed the $\ell_0$-norm and approximated it by a convex and differentiable (except at zero) sparsity promoting $\ell_1$-norm in EN.

We will discuss in the next section that in the scalable version of EN, it is more efficient to approximately solve the $\ell_0$-norm minimization problem using the greedy peeling-decoding algorithm from coding theory, which does not rely on gradient descent optimization.

EN approximately solves the following relaxed optimization problem using the SGD algorithm:

EN

$$\min_{\theta} \sum_{i=1}^{n} (y_i - g_{\theta}(\mathbf{x}_i))^2 + \alpha \, \|\mathbf{H}g_{\theta}(\mathbf{X})\|_1, \quad (7)$$

Note that despite our convex relaxation, this optimization problem is still non-convex since both the log-likelihood loss and the DNN landscape are non-convex (still differentiable) functions. In general, convergence to the global minimum can not be guaranteed due to non-convexity of DNN, however, in practice we observe that SGD converges smoothly to a useful stationary locally optimal point. To avoid convergence to locally optimal points with poor generalization performance, the DNN can be trained multiple times with several random initialization, however, as we have elaborated in the experimental section, for most of the experiments in this paper random Xavier initialization resulted in good generalization using a single initialization (no need for multiple initializations).

**Scalable Epistatic Net (EN-S).** For larger sequences (i.e., $d > 25$), the optimization algorithm in EN does not scale well with $d$. There are two factors that prevent EN from scaling to larger sequences: time and space complexity. We elaborate on these two factors. (1) In order to find the DNN landscape, we need to query the DNN $p = 2^d$ times. Regardless of how fast DNN inference is, the time complexity of this task grows exponentially with $d$. For example, it would take years to query the DNN with the simplest structure on all the binary sequences of length $d = 236$ in the avGFP protein landscape. Furthermore, finding the WH transform of the DNN landscape, even using FWHT with $\mathcal{O}(p\log p)$ computational cost, will not be possible since the computational cost grows exponentially with $d$. 2) The WH matrix $\mathbf{H}$ is a $p \times p$ matrix and the DNN landscape $g_{\theta}(\mathbf{X})$ is a $p$-dimensional vector. Regardless of the time required to find those matrices, they need exponential memory to store, which again becomes infeasible for even moderate values of $d$. We need a method that scales sublinear in $p$ (i.e., $\mathcal{O}(\text{polylog } p)$) both in time and space complexity.

Here, we develop EN-S to approximately solve our optimization problem efficiently. We first perform a change of variables and define the WH transform of the DNN landscape as $\mathbf{u} = \mathbf{H}g_{\theta}(\mathbf{X})$ and set it as an explicit constraint in the optimization problem. Following this change of variable, we reformulate the optimization problem in equation (7) as,

$$\min_{\theta, \mathbf{u}} \sum_{i=1}^{n} (y_i - g_{\theta}(\mathbf{x}_i))^2 + \alpha \, \|\mathbf{u}\|_1 \text{ subject to } \mathbf{u} = \mathbf{H}g_{\theta}(\mathbf{X}). \quad (8)$$

This change of variable enables us to use an augmented Lagrangian method to decouple the optimization problem in equation (7) into two subproblems: (1) updating the weights of DNN using SGD, and, 2) finding the WH transform of DNN using a fast greedy algorithm based on sparse-graph codes. The alternating direction method of the multipliers (ADMM) is a variant of the augmented Lagrangian methods that use partial updates for the dual variables and provides a principled framework to decouple the optimization problem above. Following the scaled-dual form of ADMM[23], we decoupled the optimization problem above into two separate minimization problems and a dual update. At iteration $t$, we first fix $\mathbf{u}_t \in \mathbb{R}^p$ and solve a $\theta$-minimization problem, then fix $\theta_t \in \mathbb{R}^p$ and solve a $\mathbf{u}$-optimization problem, and finally update the dual variable $\lambda \in \mathbb{R}^p$ as follows:

- $\theta - \text{minimization}$ $\quad \theta^{t+1} = \arg\min_{\theta} \sum_{i=1}^{n} (y_i - g_{\theta}(\mathbf{x}_i))^2 + \frac{\rho}{2}\|\mathbf{H}g_{\theta}(\mathbf{X}) - \mathbf{u}^t + \lambda^t\|_2^2$
- $\mathbf{u} - \text{minimization}$ $\quad \mathbf{u}^{t+1} = \arg\min_{\mathbf{u}} \alpha \, \|\mathbf{u}\|_1 + \frac{\rho}{2} \, \|\mathbf{H}g_{\theta}(\mathbf{X}) - \mathbf{u} + \lambda^t\|_2^2$
- dual update $\quad \lambda^{t+1} = \lambda^t + \mathbf{H}g_{\theta^{t+1}}(\mathbf{X}) - \mathbf{u}^{t+1},$

where $\rho \in \mathbb{R}$ is a hyperparameter set using cross-validation. Note that the time and space scaling issues remain here and will be addressed momentarily. Assuming an infinite time and space budget, the $\theta$-minimization problem can be tackled using SGD and the $\mathbf{u}$-minimization problem can be solved by projecting $\mathbf{w}^{t+1} := \mathbf{H}g_{\theta^{t+1}}(\mathbf{X}) + \lambda^t$ onto the $\ell_1$-norm ball of radius $\rho/\alpha$. This projection can be solved using the soft-thresholding operator in Lasso[33]:

$$\mathbf{u}_i^{t+1} = \begin{cases} \mathbf{w}_i^{t+1} - \rho/2\alpha & \text{if } \mathbf{w}_i^{t+1} > \rho/2\alpha \\ 0 & \text{if } \rho/2\alpha \leq \mathbf{w}_i^{t+1} \leq \rho/2\alpha \\ \mathbf{w}_i^{t+1} + \rho/2\alpha & \text{if } \mathbf{w}_i^{t+1} < \rho/2\alpha. \end{cases} \quad (9)$$

Unfortunately, all the three steps above still have exponential time and space scaling with $d$. In what follows we will show how to exploit the sparsity of the WH transform of the DNN landscape $\mathbf{u} = \mathbf{H}g_{\theta}(\mathbf{X})$ to reformulate new minimization steps such that we need to subsample only a logarithmic factor $\mathcal{O}(\text{polylog } p)$ of rows in $\mathbf{H}$ and approximately solve these steps in sublinear time and space

complexity in $p$ (i.e., at most polynomial in $d$). We call this regularization scheme EN-S.

The first step to arrive at the EN-S regularization scheme is to reformulate the optimizations above such that the WH matrix $\mathbf{H}$ appears as a multiplicative term behind the dual variable $\lambda$ and $\mathbf{u}$. This enables us to convert the $\mathbf{u}$-minimization problem from a $\ell_1$-norm ball projection to a sparse WH recovery problem with $\mathbf{H}$ as the basis, for which we have fast solvers from signal processing and coding theory. Note that $\|\mathbf{H}g_{\theta}(\mathbf{X}) - \mathbf{u}^t + \lambda^t\|_2^2 = \|g_{\theta}(\mathbf{X}) - \mathbf{H}\mathbf{u}^t + \mathbf{H}\lambda^t\|_2^2$ and $\|\mathbf{H}g_{\theta^{t+1}}(\mathbf{X}) - \mathbf{u} + \lambda^t\|_2^2 = \|[g_{\theta^{t+1}}(\mathbf{X}) + \mathbf{H}\lambda^t] - \mathbf{H}\mathbf{u}\|_2^2$ because $\mathbf{H}$ is a unitary matrix. Therefore, we can write the optimization steps above as

- $\theta - \text{minimization}$ $\quad \theta^{t+1} = \arg\min_{\theta} \sum_{i=1}^{n} (y_i - g_{\theta}(\mathbf{x}_i))^2 + \frac{\rho}{2}\|g_{\theta}(\mathbf{X}) - \mathbf{H}\mathbf{u}^t + \mathbf{H}\lambda^t\|_2^2$
- $\mathbf{u} - \text{minimization}$ $\quad \mathbf{u}^{t+1} = \arg\min_{\mathbf{u}} \alpha\|\mathbf{u}\|_1 + \frac{\rho}{2}\|[g_{\theta^{t+1}}(\mathbf{X}) + \mathbf{H}\lambda^t] - \mathbf{H}\mathbf{u}\|_2^2$
- dual update $\quad \mathbf{H}\lambda^{t+1} = \mathbf{H}\lambda^t + g_{\theta^{t+1}}(\mathbf{X}) - \mathbf{H}\mathbf{u}^{t+1}.$

Now, the $\mathbf{u}$-minimization problem is to find the WH transform of $g_{\theta^{t+1}}(\mathbf{X}) + \mathbf{H}\lambda^t$ with an $\ell_1$-norm sparsity prior. In order to solve this $\mathbf{u}$-minimization problem, we resort to the fast sparsity-enabled tools in signal processing and coding theory. This class of greedy algorithms solves the original $\ell_0$-norm minimization problem and finds the $k$-WH sparse landscape (for the specific value of $k$) in a time and space efficient manner ($\mathcal{O}(k \, \text{polylog } p)$, i.e., $\mathcal{O}(k \, \text{poly } d)$) using sparse-graph codes (see Supplementary Notes for an overview of these methods). To this end, we leverage subsampling of input sequences based on patterns that induce sparse-graph codes[22]. We denote the rows corresponding to these subsampled sequences as $\mathbf{X}_T$, where $|T| \sim \mathcal{O}(k \log^2 p)$. The subsampling induces a linear mixing of WH coefficients such that a belief propagation algorithm (peeling-decoding) over a sparse-graph code recovers a $p$-dimensional noisy landscape with $k$ non-zero WH coefficients in the sublinear sample (i.e., $\mathcal{O}(k \log^2 p)$) and time complexity (i.e., $\mathcal{O}(k \log^3 p)$) with high probability[13,22,24,25] (see Supplementary Materials for a full discussion). This addresses both the time and space scalability issues in solving the $\mathbf{u}$-minimization problem.

In order to resolve the time and space scalability issues in the $\theta$-minimization problem and the dual update we introduce a novel approximation. We follow the subsampling patterns dictated by the sparse-graph codes in solving the $\mathbf{u}$-minimization problem, and restrict both the $\theta$-minimization problem and the dual update to those subsamples as well to arrive at,

- $\theta - \text{minimization}$ $\quad \theta^{t+1} = \arg\min_{\theta} \sum_{i=1}^{n} (y_i - g_{\theta}(\mathbf{x}_i))^2 + \frac{\rho}{2}\|g_{\theta}(\mathbf{X}_T) - \mathbf{H}_T\mathbf{u}^t + \gamma^t\|_2^2$
- $\mathbf{u} - \text{minimization}$ $\quad \mathbf{u}^{t+1} = \arg\min_{\mathbf{u}} \alpha\|\mathbf{u}\|_0 + \frac{\rho}{2}\|[g_{\theta^{t+1}}(\mathbf{X}_T) + \gamma^t] - \mathbf{H}_T\mathbf{u}\|_2^2$
- dual update $\quad \gamma^{t+1} = \gamma^t + g_{\theta^{t+1}}(\mathbf{X}_T) - \mathbf{H}_T\mathbf{u}^{t+1},$

where $\gamma^t := \mathbf{H}_T\lambda^{t+1} \in \mathbb{R}^{|T|}$ and $\mathbf{H}_T$ comprises the rows of $\mathbf{H}$ that are in $T$. Note that the change of dual variable $\gamma^t = \mathbf{H}_T\lambda^{t+1}$ is only possible because in all the three steps the dual variable $\lambda^{t+1}$ appears in the WH basis. Note that while the columns of the subsampled WH matrix $\mathbf{H}_T$ still live in a $p$-dimensional space, this matrix is never instantiated in memory because it only appears as $\mathbf{H}_T\mathbf{u}$, where $\mathbf{u}$ is a $k$-sparse vector. Therefore, $\mathbf{H}_T\mathbf{u}$ is computed on the fly by only finding the columns of the (row-subsampled) WH matrix $\mathbf{H}_T$ that corresponds to the non-zero values in $\mathbf{u}$. The final EN-S method iterates over these three steps to train the DNN until convergence. We indicate the algorithm to solve each step in brackets:

EN-S

- $\theta - \text{minimization}$ $\quad \theta^{t+1} = \arg\min_{\theta} \sum_{i=1}^{n} (y_i - g_{\theta}(\mathbf{x}_i))^2 + \frac{\rho}{2}\|g_{\theta}(\mathbf{X}_T) - \mathbf{H}_T\mathbf{u}^t + \gamma^t\|_2^2$ [SGD]
- $\mathbf{u} - \text{minimization}$ $\quad \mathbf{u}^{t+1} = \arg\min_{\mathbf{u}} \alpha\|\mathbf{u}\|_0 + \frac{\rho}{2}\|[g_{\theta^{t+1}}(\mathbf{X}_T) + \gamma^t] - \mathbf{H}_T\mathbf{u}\|_2^2$ [Peeling]
- dual update $\quad \gamma^{t+1} = \gamma^t + g_{\theta^{t+1}}(\mathbf{X}_T) - \mathbf{H}_T\mathbf{u}^{t+1}$ [Directly computed]

All the three steps above in the EN-S method scale sublinearly with $p$ (i.e., at most polynomial with $d$) both in terms of time and space complexity.

**Experimental setup.** The architecture of DNN was selected in isolation (i.e., without any WH regularization). In our architecture search, we considered a four-layer fully connected DNN with batch normalization and leaky ReLU as the activation function. The dimension of the layers was set to $d \times fd, fd \times fd, fd \times d$, and the dimension of the final layer was $d \times 1$, where $f$ is an expansion factor. We searched for a value of $f$ that resulted in the best generalization accuracy on an independent data set—a prediction task on DNA repair landscapes[13] which we did not use for evaluation in this paper. DNN prediction performance was stable around $f = 10$ with the highest validation accuracy on the independent data set. We selected $f = 10$ in all our experiments, except for the experiments done on the avGFP landscape[2], where due to the sheer dimensionality of the problem (i.e., $d = 236$), we set $f = 1$ (on limited independent tests with $f = 10$ on the same landscape, we observed no considerable difference in prediction accuracy). The weights of the DNN were always initialized with the Xavier uniform initialization[34]. We used the exact same initialization (random seed) for the baseline DNN with and without EN(-S) regularization to ensure that we solely capture the effect of regularization and not the variations due to the initialization of DNN. We used the

Adam optimizer in all the steps of the methods requiring SGD and a learning rate of 0.001, which resulted in the best validation accuracy. We set $\alpha = 0.1$ in EN. For the DNN with EN(-S) regularization, a learning rate of 0.01 resulted in the best validation accuracy. In EN-S, the hyperparameters $\alpha$ and $\rho$ have to be jointly set since they are dependent. We set $\alpha = 1$ and $\rho = 0.01$ in EN-S although other value pairs could have resulted in the same accuracy. The validation accuracy of DNN was monitored and used for early stopping to avoid over-fitting based on the performance on a hold-out validation set (with a maximum of 1000 epochs). We used the exact same validation set to perform hyperparameter tuning of the baseline algorithms, including the Lasso family, random forest, and gradient boosted trees.

For the family of Lasso regression, we performed an extra step to improve the prediction performance. We selected the top most recovered coefficients and performed ordinary least squares (OLS) on the reduced problem. This step improves the robustness and the prediction accuracy of Lasso[35]. Therefore, in addition to the standard $\lambda$ regularization parameter, which strikes a balance between sparsity and the fidelity term (i.e., the mean squared error), we also did hyperparameter tuning for the number of top coefficients in the OLS (note that the regular Lasso is included in our hyperparameter search and appears when all the non-zero coefficients are selected to perform OLS). We did a grid search over the hyperparameter $\lambda$ and the number of top coefficients in Lasso. For $\lambda$ we considered 50 values spanning the range $[10^{-7}, 1]$. Overall, this comprised of an exhaustive hyperparameter search to make sure the best performance of Lasso is being captured.

For training gradient boosted trees and random forests baselines, we used packages from sklearn in python. We did hyperparameter tuning for max depth and the number of estimators, using the default values for all other parameters. For max depth, we considered parameters ranging from 1 to a constant times the maximum number of mutations in the fitness function (i.e., $d$), for the number of estimators we considered values in $\{10, 50, 100, 200, 300, 400, 500, 1000, 2000, 3000\}$, and chose the pair that resulted in best validation accuracy. As a general trend, we observed that larger numbers of estimators result in higher validation accuracies before they saturate.

Here, we report the hyperparameters that resulted in the highest validation accuracy, that is, the ones we selected in our experiments. For the avGFP landscape, we set the number of estimators to 300 and max depth to 11 for gradient boosted trees and set the number of estimators to 100 and max depth to 55 for random forests. We set $\lambda = 1 \times 10^{-4}$ for Lasso regression when considering up to first-order interactions and $\lambda = 1 \times 8^{-4}$ when considering up to second-order interactions. For the GB1 landscape, we set the number of estimators to 100 and max depth to 2 for both gradient boosted trees and random forests. We set $\lambda = 7 \times 10^{-3}$ for Lasso regression when considering up to first-order interactions and $\lambda = 2.5 \times 10^{-2}$ when considering up to second-order interactions. For the protein landscape in Fig. 3, we set the number of estimators to 3000 and the max depth varied between the values in the sets $\{1, 2, 3, 4\}$ and $\{1, 2, \ldots, 15\}$ across the random repeats of the experiments with different train, test, and validation set, respectively for gradient boosted trees and random forest; the value with the best validation performance was selected for each repeat. For the bacterial landscapes in Fig. 2, we set the number of estimators to 300 and the max depth varied between the values in the set $\{1, 2, 3\}$ across the random repeats of the experiments with different train, test, and validation set; the value with the best validation performance was selected for each repeat.

In all the relevant protein and biological data sets, we performed a two-sided T-test for the null hypothesis that the independent prediction from DNN with and without EN regularization (across random Xavier initialization) has identical average (expected) values and reported the $p$-values.

**Prepossessing the fitness landscapes.** For some of the landscapes tested in this paper, we followed the Box-Cox power transform method as described in ref. [18] to remove possible global nonlinearities from the landscape. Although the effect of removing such nonlinearities was small in our analysis, global nonlinearities in general can produce high-order epistatic interactions that are not truly needed. Removing these nonlinearities can reduce noise and increase epistatic sparsity. Nevertheless, one can completely ignore this preprocessing step and rely on DNN with EN regularization to capture the global nonlinearites and infer the fitness landscape for prediction purposes.

**Reporting summary.** Further information on research design is available in the Nature Research Reporting Summary linked to this article.

## Data availability

The canonical bacterial fitness data used in Fig. 2 are available in the github repository associated with the Ref. [18]. The *E. quadricolor* fluorescent protein data used in Fig. 3 are available in Supplementary Data 3 of https://doi.org/10.1038/s41467-019-12130-8 Ref. [3]. The avGFP protein data used in Fig. 4 are available in the figshare data repository under accession code 3102154. The GB1 protein data used in Fig. 4 are available in Supplementary file 1 and Supplementary file 2 of https://elifesciences.org/articles/16965/figures Ref. [1]. All other data generated or analyzed in this study are included in this published article (and in its accompanying Supplementary Information and Supplementary Data).

## Code availability

A software for the Epistatic Net regularization algorithms has been developed in Python and is publicly available in our github repository at https://github.com/amirmohan/epistatic-net[36].

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

## Acknowledgements

A.A., O.O., and K.R. were supported by the NSF (1703678) and ARO (W911NF2110117). H.N. was supported by the National Library of Medicine of the NIH (T32LM012417); the content is solely the responsibility of the author and does not necessarily represent the official views of the NIH. O.O.K. was supported by the NSF (1748692). D.H.B and J.L. were supported by the DOE, Office of Biological and Environmental Research, Genomic Science Program Lawrence Livermore National Laboratory's Secure Biosystems Design Scientific Focus Area (SCW1710). The authors thank Clara Wong-Fannjiang for insightful discussions.

## Author contributions

A.A., H.N., and O.O. designed research; A.A., H.N., O.O., D.B., and Y.H. conducted research; A.A., O.O., O.O.K., J.L., and K.R. wrote the manuscript.

## Competing interests

Jennifer Listgarten is on the Scientific Advisory Board for Foresite Labs and Patch Biosciences. Other authors declare no competing interests.
