## [Peer Review File · Nature Communications]

Epistatic Net Allows the Sparse Spectral Regularization of Deep Neural Networks for Inferring Fitness FunctionsReviewers' Comments:

Reviewer #1:

Remarks to the Author:

This manuscript describes an elegant computational method to detect protein residue-residue interactions that may explain fitness functions by combining deep neural network (DNN) and sparse code recovery. The authors tested their method on 4 examples and showed that the sparsity regularized DNN indeed works and produces better results than existing methods and non-sparse DNN. Overall, the method presented in this manuscript is very interesting and may be applied to other related problems. The results also look good. Nevertheless the paper also has some issues:

- 1) Deeper result analysis is needed. In addition to show that their method may result in better prediction accuracy, the authors shall also analyze those identified first-order, 2nd-order and higher-order interactions to check if they are biologically meaningful. In particular, what insight can be gained if we map the identified interactions to the protein structure? Do they correspond to spatially proximate residues?
- 2) In Fig 2c, looks like the top 3 identified interactions are 00010, 00100 and 10000, all of which correspond to only single-position information. Maybe the authors shall give some explanation about this. Is it possible that some important 2nd-order interactions are missed by the proposed method?
- 3) In Fig. 3, the prediction accuracy for larger number (>100) of sequences shall also be displayed. In particular, I would like to know in what condition all the competing methods may have similar performance.
- 4) Please use plain language to explain the main idea of the sparse graph coding component in Fig 1c so that the proposed method is accessible to more readers. That is, how to recover the top-k WH coefficient from the DNN output ?

Reviewer #2:

Remarks to the Author:

Aghazadeh et al. report aims to solve nonlinearity in epistatic fitness functions with a deep learning approach combined with sparse regularization. The authors cleverly identified the sparse nature of these functions and introduced a network that can impose this biologically relevant constraint in a tractable manner. As deep learning in biological contexts matures, domain specific knowledge applied to network architectures can help reduce training effort, and this manuscript demonstrates a critical step in that approach. In that vein, the authors show that through their method of WH transformation, smaller training sizes can achieve similar predictive results.

However, the gains in predictive accuracy are only minimal. In the experiments where such an approach can be transformative (e.g. identifying epistatic sites across several residues in a protein), the absolute increase in accuracy is not noteworthy (E. quadricolor FP, avGFP, GB1) when compared to a baseline DNN. The authors do demonstrate prediction improvements when analyzing 5 mutational sites in organism specific fitness analyses, however these fitness functions are only described by 32 terms and do not translate to a utility where thousands of genes in a genome can be analyzed. Furthermore, there is lack of biological context in describing the results that can perhaps provide insight into what the model is learning and drive deeper understanding of epistasis (i.e. which types of residues are being correctly predicted vs not, how does distance of epistatic residues affect network performance, etc.). While I do not doubt the potential of such an approach particularly for genome/protein wide epistasis, it was not adequately demonstrated here.

Comments:

1. In Figure 1, it is unclear to what the circled numbers are referencing.
2. The GB1 analysis is inherently flawed. Sparsity is being promoted across $d=80$, however, certain amino acid mutations are biologically impossible (i.e. cannot have multiple mutations in the same site).
3. While prediction accuracy is increased in every analysis, it is less pronounced in the protein level studies (E. quadricolor FP, avGFP, GB1), where, notably, d is increased. Additional discussion or analysis is needed to parse this effect.
4. Epistasis can be either positive or negative. Does the model preferentially predict one vs the other?
5. In a similar vein, all of the analysis is done at a macro level, but some model insights can be gleaned from looking at micro trends. Does the model tend to predict certain residues or distance between residues in the protein fitness analysis? Can any meaningful conclusion be drawn from the top or lowest scoring coefficients?
6. The R^2 value for E. quadricolor FP, avGFP, and GB1 are presented without an accompanying scatter plot and should be provided in the supplement.

RESPONSE TO REVIEWS

Sparse Epistatic Regularization of Deep Neural Networks for Inferring Fitness Functions

Amirali Aghazadeh, Hunter Nisonoff, Orhan Ocal, David H. Brookes, Yijie Huang,
O. Ozan Koyluoglu, Jennifer Listgarten, and Kannan Ramchandran

University of California, Berkeley

May 17, 2021

We would like to thank the referees for reviewing our manuscript and providing such thoughtful comments. We have addressed each referee comment and have submitted a revised manuscript. We have highlighted the changes made in the text using the color blue. Below, we discuss each referee comment in detail.

Reviewer #1 (*Expertise: Predicting protein function from sequence*):

This manuscript describes an elegant computational method to detect protein residue-residue interactions that may explain fitness functions by combining deep neural network (DNN) and sparse code recovery. The authors tested their method on 4 examples and showed that the sparsity regularized DNN indeed works and produces better results than existing methods and non-sparse DNN. Overall, the method presented in this manuscript is very interesting and may be applied to other related problems. The results also look good. Nevertheless the paper also has some issues:

1) Deeper result analysis is needed. In addition to show that their method may result in better prediction accuracy, the authors shall also analyze those identified first-order, 2nd-order and higher-order interactions to check if they are biologically meaningful. In particular, what insight can be gained if we map the identified interactions to the protein structure? Do they correspond to spatially proximate residues?

We have conducted new systematic analysis where we compare the quality of recovered (low and high-order) epistatic interactions in terms of their biological relevance in the protein function prediction task. We evaluate the error in finding the experimentally measured epistatic interactions using DNN with and without the EN regularization and compare them with the other competing machine learning algorithms. In Figure 3b of the revised manuscript (also illustrated below), we have quantified the epistatic recovery error in terms of the normalized mean squared error (NMSE) for the combinatorial landscape of *Entacmaea quadricolor* fluorescent protein with 13 mutations and a total of 8192 possible

interactions; which is the largest biological landscape for which we know the ground truth epistatic interactions.

Our analysis shows that DNN with EN regularization recovers the epistatic interactions with significantly lower NMSE among all the training sizes ($\Delta\text{NMSE} > 0.07$, $P < 9e-5$), and has consistently lower epistatic recovery error compared to the rest of the competing baselines.

Regarding the suggested comparisons of the recovered higher order interactions to the 3D structure of the protein, we emphasize that the focus of this paper is on predicting the protein function rather than structure. While we expect some degrees of correlation between the 3D structure of the protein and the epistatic interactions, the connection is not completely clear in most cases and in fact motivates the development of predictive tools that map directly from sequence to function. The WH transform of the combinatorial landscape (if available) reveals the biologically meaningful interactions between the residues that explain the underlying property of the protein. This is why in the analysis stated above we have compared the recovered epistatic interaction to those ground truth biological interactions.

Regardless, in order to fully address reviewer's comments and in addition to the epistatic analysis above, we have compared the recovered second-order and third-order epistatic interactions of the DNN with (Figure S6b) and without (Figure S6c) EN regularization to the

contact map of the *Entacmaea quadricolor* fluorescent protein (Figure S6a). Both models are trained on the same random subset of $m=60$ labeled proteins (out of 8192).

Figure above shows the prediction performance and epistatic recovery of DNN with and without EN regularization and figure below shows the performance of the corresponding DNNs in predicting the contact map.

d

F1 score	DNN	DNN + EN
2 nd order contact	0.68	0.76
3 rd order contact	0.66	0.67

Our analysis demonstrates that even though the final objective of both the regularized and unregularized DNNs is predicting the protein function and not predicting the contact map, the epistatic interactions of DNNs give some information about the contacts in the 3D structure of the protein. In particular, DNN with EN regularization shows lower false positive rate in predicting the contact map compared to the unregularized DNN, which results in a higher F_1 score (i.e., the harmonic mean of the precision and recall rates). Table

above tabulates the F_1 score in recovering both the second order contact (i.e., pairs of residues with smaller than 4.5Å distance [26]) and third order contact (i.e., groups of three residues with smaller than 4.5Å pairwise distances).

We have added the scatter plots which show the prediction performance of DNNs to the main text and added the analysis of the protein contact map to the Supplementary Materials and summarized the results in lines 175-198 of the revised manuscript as follows:

“We varied the training set size from a minimum of $n=20$ proteins to a maximum of $n=100$ proteins and evaluated the accuracy of the models in 1) predicting fitness in Figure 3a in terms of R^2 and 2) recovering the experimentally measured epistatic interactions in Figure 3b in terms of normalized mean squared error (NMSE).

DNN with EN regularization significantly outperforms DNN without EN regularization in terms of prediction accuracy ($\Delta R^2 > 0.1$, $P < 10^{-5}$), consistently across all training sizes. Moreover, DNN with EN regularization recovers the experimentally measured epistatic interactions with significantly lower error ($\Delta \text{NMSE} > 0.07$, $P < 9 \times 10^{-5}$), consistently across all training sizes. Applying various forms of ℓ_1 and ℓ_2 -norm regularization on the weights of different layers of the DNN does not change the performance gap between DNN with and without EN regularization (see Figure S5 in Supplementary Materials).

...

Our analysis also reveals the improved performance of the epistatic interactions recovered by DNN with EN regularization in predicting the pairwise contacts (residues with smaller than 4.5 Å distance [26]) and triplet contacts (group of three residues with smaller than 4.5 Å pairwise distances) in the 3D structure of the protein---even though the networks are not trained for protein structure prediction task. DNN with EN regularization predicts contacts with $F_1^{\text{order-2}}=0.76$ and $F_1^{\text{order-3}}=0.68$ compared to DNN without EN regularization with $F_1^{\text{order-2}}=0.67$ and $F_1^{\text{order-3}}=0.66$ (F_1 score takes the harmonic mean of the precision and recall rates).”

2) In Fig 2c, looks like the top 3 identified interactions are 00010, 00100 and 10000, all of which correspond to only single-position information. Maybe the authors shall give some explanation about this. Is it possible that some important 2nd-order interactions are missed by the proposed method?

In our EN regularization method the threshold for accepting higher-order epistatic interactions is controlled using the hyper-parameter alpha (see equation (7)) which is set using cross validation. Alpha strikes a balance between the fidelity to the log-likelihood loss and the sparse WH regularization term. When the number of labeled data points is small, it is natural to assume that cross validation would select a larger value for alpha which results in selecting the top most epistatic interactions. In other words, the EN regularization could have accepted a larger number of higher-order interactions using a smaller value for alpha, however, that would result in accepting a number of false epistatic interactions which would compromise the overall prediction accuracy (and epistatic recovery) of the algorithm in this particular example.

In the experiments of Figure 2c, besides the three first-order interactions, DNN with EN regularization finds higher-order interactions such as 01100 which in fact match with the higher-order interactions of the measured landscapes. EN regularization finds those higher-order interactions without accepting false epistatic interactions such as 01010. Compare this with the DNN without EN regularization or the gradient boosted trees which both learn functions whose WH transform show several false epistatic interactions. We have clarified this in line 159-161 of the paper as follows:

“... The interactions recovered by DNN with EN regularization closely matches the epistatic interactions of the measured E. coli fitness function ($R^2=0.67$), a considerable improvement over DNN without EN regularization ($R^2=0.41$). EN regularization effectively “denoises” the WH spectrum of DNN by removing spurious higher-order interactions; nevertheless, given a larger training set, EN would have accepted a larger number of higher-order interactions. The WH coefficients of gradient boosted trees ($R^2=0.51$) and random forests ($R^2=0.36$) also shows several spurious high-order interactions. ... ”

3) In Fig. 3, the prediction accuracy for larger number (>100) of sequences shall also be displayed. In particular, I would like to know in what condition all the competing methods may have similar performance.

We want to clarify that the maximum value of 100 labeled protein sequences that we have considered in Figure 3 for our comparisons amounts to more than 1% of all the possible combinatorial sequences. In practice, the number of assay-labeled sequences available is much smaller than the size of the entire combinatorial space of sequences (10^{66} times smaller for the avGFP protein landscape of Sarkisyan *et al.* [2]) and this is the reason why in this work we have focused on developing a regularization scheme that improves the performance of DNNs in a small sample regime.

Regardless, in order to address reviewer’s comment we have extended both the prediction accuracy plot (Figure 3a) as well as the newly added epistatic recovery error plot (Figure 3b) to include up to two times larger number of training proteins (i.e., up to 2.5% of the total landscape), illustrated below:

Our analysis shows that DNN with EN regularization consistently outperforms DNN without EN regularization and other competing baselines even when we consider larger than 1% of all the landscape in the training set. Naturally, the regularization power of EN reduces with larger training sizes, however, we still observe a performance gap. We have added the plots above to the Supplementary Materials, keeping the focus of Figure 3 on low-sample regimes, which is the key focus in protein function prediction. We have added this discussion to the main text to elaborate on the behaviour of the regularization for larger sample sizes as suggested by the reviewer:

“... The performance gap naturally reduces for larger training sets, however, it stays consistently positive even up to $n=200$ (i.e., 2.5% of the entire combinatorial landscape), which is typically larger than the number of available labeled sequences in protein function prediction problems (see Figure S5 in Supplementary Materials). ... ”

4) Please use plain language to explain the main idea of the sparse graph coding component in Fig 1c so that the proposed method is accessible to more readers. That is, how to recover the top-k WH coefficient from the DNN output ?

We have stated the details of the algorithm in the Supplementary Materials (see Sections 7.1, 7.2, and 7.3). Briefly, the main procedure behind using sparse graph codes to recover the top-k WH coefficients is: 1) subsampling the DNN uniformly, 2) performing smaller WH transforms (which are computationally cheaper than full WH transform), and 3) conducting sparse recovery on the induced sparse bipartite graph (as the result of the subsampling) using a greedy peeling algorithm. Subsampling a signal (in this case the DNN landscape) is known to create aliasing patterns that in general corrupt the signal. However, in this work we subsample the DNN in a smart way such that after taking smaller WH transforms the sparse recovery problem can be cast as finding the location of a few number of non-zero nodes in a sparse bipartite graph (see Figure S1). The peeling algorithm that we use to solve this problem is inspired by and developed in the low density parity check (LDPC) codes [22]. It identifies the nodes on the induced sparse-graph code that are connected to only a single WH coefficient and peels off the edges connected to those nodes and their contributions on the overall graph. The algorithm repeats these steps until all the edges are removed. We have further clarified the peeling algorithm using sparse-graph codes in lines 121-124 of the revised manuscript:

“... . In order to solve the first subproblem, we designed a careful subsampling of the input sequence space [22] that induces a linear mixing of the WH coefficients such that a greedy belief propagation algorithm (peeling-decoding) over a sparse-graph code recovers the noisy DNN landscape in sublinear sample (i.e., $O(k \log^2 p)$) and time (i.e., $O(k \log^3 p)$) complexity in p (with high probability) [13,22,24,25]. Briefly, the peeling-decoding algorithm identifies the nodes on the induced sparse-graph code that are connected to a single WH coefficient and peels off the edges connected to those nodes and their contributions on the overall graph. The algorithm repeats these steps until all the edges are removed. We solved the second subproblem using the SGD algorithm. EN-S alternates between these two steps until convergence (see Methods for more detail).”

Reviewer #2 (Expertise: Predicting protein function from sequence):

Aghazadeh et al. report aims to solve nonlinearity in epistatic fitness functions with a deep learning approach combined with sparse regularization. The authors cleverly identified the sparse nature of these functions and introduced a network that can impose this biologically relevant constraint in a tractable manner. As deep learning in biological contexts matures, domain specific knowledge applied to network architectures can help reduce training effort, and this manuscript demonstrates a critical step in that approach. In that vein, the authors show that through their method of WH transformation, smaller training sizes can achieve similar predictive results.

However, the gains in predictive accuracy are only minimal. In the experiments where such an approach can be transformative (e.g. identifying epistatic sites across several residues in a protein), the absolute increase in accuracy is not noteworthy (E. quadricolor FP, avGFP, GB1) when compared to a baseline DNN.

The authors do demonstrate prediction improvements when analyzing 5 mutational sites in organism specific fitness analyses, however these fitness functions are only described by 32 terms and do not translate to a utility where thousands of genes in a genome can be analyzed. Furthermore, there is lack of biological context in describing the results that can perhaps provide insight into what the model is learning and drive deeper understanding of epistasis (i.e. which types of residues are being correctly predicted vs not, how does distance of epistatic residues affect network performance, etc.). While I do not doubt the potential of such an approach particularly for genome/protein wide epistasis, it was not adequately demonstrated here.

In the revised manuscript, we have conducted new systematic analysis where we compared the quality of recovered epistatic interactions in terms of their biological relevance in the protein function prediction task. In addition, we conducted comparisons of the recovered epistatic interactions to the 3D structure of the protein---even though the objective of the paper is on protein function prediction rather than structure. All these analyses demonstrate the superior performance of DNN with EN regularization in finding biologically relevant epistatic interactions which also have higher predictive power compared to DNN without EN regularization. We have also revised the manuscript to better reflect the significance of the performance gap (in terms of absolute gain in prediction accuracy and epistatic recovery along with their corresponding p-values) in all the biological landscapes suggested by the reviewer including the protein landscapes. We have detailed the results of these analyses below in response to each of the comments.

Comments:

1. In Figure 1, it is unclear to what the circled numbers are referencing.

The circle with number 1 refers to the log-likelihood loss, the circle with number 2 refers to the exact WH loss, and the circle with number 3 refers to the approximate WH loss recovered by the peeling algorithm. In order to avoid any confusion, we have removed the circles and explicitly stated what we mean by each circle in the revised version of Figure 1 as illustrated below.

2. The GB1 analysis is inherently flawed. Sparsity is being promoted across $d=80$, however, certain amino acid mutations are biologically impossible (i.e cannot have multiple mutations in the same site).

We do understand that certain interaction terms in the binary encoding of the sequence correspond to biologically impossible inputs; however, that does not detract from the analysis. We clarify that those biologically implausible interactions exist in the WHT of DNN without the EN regularization. Our method simply encourages those biologically implausible terms to be zero; however, one could also apply the sparse regularization only on the biologically plausible terms. Regardless, the algorithm developed in this work could be coupled with other non-binary inputs; however, we emphasize that there is nothing inherently flawed with representing the proteins as a binary vector and regularizing sparsity in that domain. We have revised lines 250-259 on the details of GB1 analysis which might have caused the confusion, as follows:

“... . EN-S subsamples DNN at 215,040 proteins in order to perform the sparse epistatic regularization, which is about 10^{18} times smaller than the entire sequence space. Despite such an enormous level of undersampling, the DNN regularized with EN-S consistently outperforms the competing baselines and the EN-S unregularized DNN ($\Delta R^2 > 0.035$, $P < 0.05$, see Figure S9 for the corresponding scatter plots). The performance gap between the DNNs with and without EN-S regularization is naturally smaller compared to the same gap in

the Entacmaea quadricolor fluorescent protein landscape. This is because the protein landscape of Entacmaea quadricolor is defined over 13 mutational sites (with 8192 possible positional interactions and two possible amino acids for each site) while the protein landscape of GB1 is defined over 4 mutational sites (with 16 possible positional interactions and 20 possible amino acids for each site); the former benefits more from promoting sparsity among a larger number of (biologically-meaningful) epistatic positional interactions. ...”

3. While prediction accuracy is increased in every analysis, it is less pronounced in the protein level studies (E. quadricolor FP, avGFP, GB1), where, notably, d is increased. Additional discussion or analysis is needed to parse this effect.

We want to highlight the significance of the results of our regularization algorithm on the protein data sets. In the case of *E. quadricolor* FP, DNN with EN regularization, outperforms the baseline DNN without the regularization with 0.1 point increase in terms of R^2 (p-value < 1e-5) and 0.07 point decrease in terms of NMSE in epistatic recovery (p-value < 9e-5) consistently across all the training sizes (see the revised version of Figure 3a and our response to the next comment). To put this into perspective, in order for the DNN baseline to compensate for such a performance gap, it would require up to three times more number of labeled sequences. We have clarified this in the line 179 of the revised manuscript as follows:

“... . DNN with EN regularization significantly outperforms DNN without EN regularization in terms of prediction accuracy ($\Delta R^2 > 0.1$, $P < 10^{-5}$), consistently across all training sizes. ...”

We have discussed in the manuscript the reasons why the performance gap is less pronounced for the avGFP protein. We attribute this to the local properties of the avGFP landscape around the wild type protein rather than a behaviour intrinsic to the EN regularization. The landscape of avGFP in the neighbourhood with a small (<5) Hamming distance from the wild type is fairly low-order and mostly predictive using low-order interactions. This has been discussed in line 232 of the manuscript as stated below:

“... . We speculate that this is due to the nature of the local landscape of avGFP around the wild type protein, where most of the variance can be explained by first-order interactions and the rest can be explained by higher-order interactions that are spread throughout the WH spectrum. ...”

Even in the presence of the overall low-order intrinsic properties, we have done additional analysis (shown below) to demonstrate the power of our algorithm in recovering the higher-order interactions in the massively combinatorial landscape of avGFP. We plot the histogram of the order of epistatic interactions recovered by the peeling algorithm while training the DNN using the EN regularization, as well as the gain in prediction accuracy after adding those higher-order interactions to a purely linear model. The analysis reveals that the performance gap between DNNs with and without EN regularization can be explained by the recovered higher order interactions. *Note that finding these interactions is not computationally feasible using conventional statistical tools such as the Lasso algorithm.*

We have added these plots to Figure 4 and revised the manuscript as follows:

“... . Figure 4d illustrates the histogram of the order of epistatic interactions recovered by invoking the peeling algorithm in every iteration of the EN-S regularization scheme. Figure 4e depicts the gain in prediction accuracy after adding the recovered interactions to a purely linear model, suggesting that the difference in prediction accuracy of DNN with and without regularization can be explained (approximately) by a collection of large number of WH coefficients with small magnitude---this analysis further demonstrates the computational power of EN-S in recovering higher-order interactions in such massively large combinatorial space of interactions. ...”

4. Epistasis can be either positive or negative. Does the model preferentially predict one vs the other?

The model does not preferentially predict positive or negative epistasis. The EN regularization is completely agnostic to sign as it promotes sparsity among epistatic interactions based on their magnitude. We have clarified this in line 103 of the revised manuscript:

“... . EN takes the WH transform of the DNN landscape and adds the sparsity-promoting ℓ_1 -norm (i.e., the sum of the absolute values) of the WH coefficients (or total sum of the magnitude of epistasis) to the log-likelihood loss. ...”

5. In a similar vein, all of the analysis is done at a macro level, but some model insights can be gleaned from looking at micro trends. Does the model tend to predict certain residues or distance between residues in the protein fitness analysis? Can any meaningful conclusion be drawn from the top or lowest scoring coefficients?

We have conducted new systematic analysis where we compare the quality of recovered (low and high-order) epistatic interactions in terms of their biological relevance in the protein function prediction task. We evaluate the error in finding the experimentally measured epistatic interactions using DNN with and without the EN regularization and compare them with the other competing machine learning algorithms. In Figure 3b of the revised manuscript (also illustrated below), we have quantified the epistatic recovery error in terms of the normalized mean squared error (NMSE) for the combinatorial landscape of *Entacmaea quadricolor* fluorescent protein with 13 mutations and a total of 8192 possible interactions; which is the largest biological landscape for which we know the ground truth epistatic interactions.

Our analysis shows that DNN with EN regularization recovers the epistatic interactions with significantly lower NMSE among all the training sizes ($\Delta\text{NMSE} > 0.07$, $P < 9e-5$) and

has consistently lower epistatic recovery error compared to the rest of the competing baselines.

Regarding the suggested comparisons of the recovered higher order interactions to the 3D structure of the protein, we emphasize that the focus of this paper is on predicting the protein function rather than structure. While we expect some degrees of correlation between the 3D structure of the protein and the epistatic interactions, the connection is not completely clear in most cases and in fact motivates the development of predictive tools that map directly from sequence to function. The WH transform of the combinatorial landscape (if available) reveals the most biologically relevant (higher-order) interactions between the residues that explain the underlying property of the protein. This is why in the analysis stated above we have compared the recovered epistatic interaction to those ground truth biological interactions.

Regardless, in order to fully address reviewer's comments and in addition to the epistatic analysis above, we have compared the recovered second-order and third-order epistatic interactions of the DNN with (Figure S6b) and without EN regularization (Figure S6c) to the contact map of the *Entacmaea quadricolor* fluorescent protein (Figure S6a). Both models are trained on the same subset of $m=60$ labeled proteins (out of 8192).

Figure above shows the prediction performance and epistatic recovery of DNN with and without EN regularization and figure below shows the performance of the corresponding DNNs in predicting the contact map.

Our analysis demonstrates that even though the final objective of both the regularized and unregularized DNNs is predicting the protein function and not predicting the contact map, the epistatic interactions of DNNs give some information about the contacts in the 3D structure of the protein. In particular, DNN with EN regularization shows lower false positive rate in predicting the contact map compared to the unregularized DNN with the same true positive, which results in a higher F_1 score (i.e., the harmonic mean of the precision and recall rates). Table above tabulates the F_1 score in recovering both the second order contact (i.e., pairs of residues with smaller than 4.5\AA distance [26]) and third order contact (i.e., groups of three residues with smaller than 4.5\AA pairwise distances).

We have added the scatter plots above showing the prediction performance of DNNs to the main text and added the analysis of the protein contact map to the Supplementary Materials and summarized the results in lines 175-198 of the revised manuscript as follows:

“We varied the training set size from a minimum of $n=20$ proteins to a maximum of $n=100$ proteins and evaluated the accuracy of the models in 1) predicting fitness in Figure 3a in terms of R^2 and 2) recovering the experimentally measured epistatic interactions in Figure 3b in terms of normalized mean squared error (NMSE).

DNN with EN regularization significantly outperforms DNN without EN regularization in terms of prediction accuracy ($\Delta R^2 > 0.1$, $P < 10^{-5}$), consistently across all training sizes. Moreover, DNN with EN regularization recovers the

experimentally measured epistatic interactions with significantly lower error ($\Delta\text{NMSE} > 0.07$, $P < 9 \times 10^{-5}$), consistently across all training sizes. Applying various forms of ℓ_1 and ℓ_2 -norm regularization on the weights of different layers of the DNN does not change the performance gap between DNN with and without EN regularization (see Figure S5 in Supplementary Materials).

...

Our analysis also reveals the improved performance of the epistatic interactions recovered by DNN with EN regularization in predicting the pairwise contacts (residues with smaller than 4.5 Å distance [26]) and triplet contacts (group of three residues with smaller than 4.5 Å pairwise distances) in the 3D structure of the protein---even though the networks are not trained for protein structure prediction task. DNN with EN regularization predicts contacts with $F_1^{\text{order } 2} = 0.76$ and $F_1^{\text{order } 3} = 0.68$ compared to DNN without EN regularization with $F_1^{\text{order } 2} = 0.67$ and $F_1^{\text{order } 3} = 0.66$ (F_1 score takes the harmonic mean of the precision and recall rates).”

6. The R^2 value for *E. quadricolor* FP, avGFP, and GB1 are presented without an accompanying scatter plot and should be provided in the supplement.

We have added the scatter plot of the predictions and the recovered epistatic interaction with and without the EN regularization for the *E. quadricolor* protein in Figures 3d, 3e, and 3f of the revised manuscript (also shown below):

We have added the corresponding scatter plots for the rest of the protein landscapes to the Supplementary Materials.

Reviewers' Comments:

Reviewer #1:

Remarks to the Author:

The authors have revised the manuscript greatly and addressed my concerns. I am satisfied with the revision.

Reviewer #2:

Remarks to the Author:

I am satisfied with the author's revisions pertaining to my comments. The effort to provide further biological context through contact maps and higher order epistatic interactions addresses my concern adequately. Furthermore, the superior performance of EN regularization is more clearly demonstrated in the manuscript. I recommend for publication.

One minor point: the color scales for the predicted contact maps should be the same in Figure S6b and S6c.

RESPONSE TO REVIEWS

Epistatic Net Allows the Sparse Spectral Regularization of Deep Neural Networks for Inferring Fitness Functions

Amirali Aghazadeh, Hunter Nisonoff, Orhan Ocal, David H. Brookes, Yijie Huang,
O. Ozan Koyluoglu, Jennifer Listgarten, and Kannan Ramchandran

University of California, Berkeley

July 17, 2021

We would like to thank the referees again for reviewing our manuscript and providing such thoughtful comments. We have addressed each referee comment and have submitted a revised manuscript. We have highlighted the changes made in the text using the color blue. Below, we discuss each referee comment in detail.

Reviewer #1 (Remarks to the Author):

The authors have revised the manuscript greatly and addressed my concerns. I am satisfied with the revision.

Reviewer #2 (Remarks to the Author):

I am satisfied with the author's revisions pertaining to my comments. The effort to provide further biological context through contact maps and higher order epistatic interactions addresses my concern adequately. Furthermore, the superior performance of EN regularization is more clearly demonstrated in the manuscript. I recommend for publication.

One minor point: the color scales for the predicted contact maps should be the same in Figure S6b and S6c.

We have modified the color scale of the contact maps accordingly.